# Rank-Based Psychological Characteristics in Brazilian Jiu-Jitsu Athletes: Mental Strength, Resilience, Grit, Self-Efficacy, Self-Control, Aggression, Life Satisfaction, and Mental Health

**DOI:** 10.3390/jfmk10020100

**Published:** 2025-03-22

**Authors:** Leandro de Lorenco-Lima, Stacey A. Gaines, Elisabeth M. Waterbury

**Affiliations:** Department of Psychology, Liberty University, Lynchburg, VA 24515, USA; sgaines18@liberty.edu (S.A.G.); emspratto@liberty.edu (E.M.W.)

**Keywords:** BJJ, martial arts, combat sports, sport psychology

## Abstract

Brazilian jiu-jitsu engagement has been found to positively impact psychological variables in children, adolescents, and adult athletes. Psychological variabilities have previously been shown among belt ranks in Brazilian jiu-jitsu athletes. **Background/Objectives:** This study aimed to explore the differences in mental strength, resilience, grit, self-efficacy, self-control, aggression, life satisfaction, and mental health disorders among the Brazilian jiu-jitsu belt ranks. **Methods:** A sample of 420 Brazilian jiu-jitsu athletes (78.8% male), including 121 white belts, 118 blue belts, 78 purple belts, 46 brown belts, and 57 black belts, between 18 and 60 years of age, completed the Mental Strength Scale, Brief Resilience Scale, Grit Scale, General Self-Efficacy Scale, Brief Self-Control Scale, Brief Aggression Questionnaire, Satisfaction with Life Scale, and Mental Health Disorders Screening Instrument for Athletes. **Results:** Results showed that Brazilian jiu-jitsu black belts presented significantly higher mental strength, resilience, self-efficacy, self-control, life satisfaction, and lower mental health disorders than white belts. No significant differences were found between white and black belts in aggression (total, physical aggression, anger, verbal aggression, and hostility). Brazilian jiu-jitsu training experience positively correlated (small effect) with mental strength, resilience, grit, self-efficacy, self-control, physical and verbal aggression, and life satisfaction. **Conclusions:** In conclusion, the results are suggestive that Brazilian jiu-jitsu black belts are generally more likely to present higher mental strength, resilience, grit, self-efficacy, self-control, life satisfaction, and better mental health than white belts, with no differences in aggression. More experienced Brazilian jiu-jitsu athletes were more likely to present higher mental strength, resilience, grit, self-efficacy, self-control, physical and verbal aggression, and life satisfaction than less experienced athletes. Future studies are encouraged to investigate whether the effects found in the current study would remain after controlling for the athlete’s age.

## 1. Introduction

As a combat sport or modern martial art, Brazilian jiu-jitsu is a highly popular grappling style among adults in the United States, with rising popularity worldwide over the last decades [1,2]. Brazilian jiu-jitsu matches emphasize taking an opponent to the ground via takedowns, throws, or pulling guard, controlling the opponent, and performing submissions such as joint locks and strangles [3]. During Brazilian jiu-jitsu matches, two athletes battle one another in quest for victory, which is determined by submission, points, loss of consciousness, or disqualification [4,5]. This inherent characteristic of Brazilian jiu-jitsu matches makes failure an unavoidable obstacle for athletes based on the very nature of two individuals competing against one another [3]. Certain psychological variables, such as the athletes’ ability to persist, are critical due to the vigorous and physical nature of matches [6] and the numerous defeats and obstacles athletes face during training [2]. These characteristics demand certain psychological skills from athletes, or at least, the willingness to develop them [2].

In general, combat sports and martial arts have previously been found to positively impact several psychological traits and skills, including mental strength [7], resilience [8,9,10], grit [11,12,13], self-efficacy [8,9,14], self-control [15,16,17,18], aggression [14,15,16,17,19,20,21], life satisfaction [17,22,23,24,25], and mental health [26,27].

The limited available psychology-related Brazilian jiu-jitsu literature has found it to be an effective intervention to decrease post-traumatic stress disorder symptoms [26,27], psychopathology symptoms [26], emotional symptoms, externalizing problems, hyperactivity/inattention, total difficulties [28], and aggression, and to increase self-control [15]. Previous studies have shown the positive impact of combat sports experience on mental strength [7], resilience [29,30,31], grit [13], self-efficacy [32], verbal aggression, suspicion, negativism, irritability, and assault [20], and life satisfaction [23]. Due to the harsh and vigorous nature of Brazilian jiu-jitsu training and matches, and the high number of obstacles faced by beginners, many athletes prematurely exit the sport [3], making it necessary to better understand the potential differences in psychological characteristics among Brazilian jiu-jitsu athletes of different belt ranks and experience levels. Moreover, the short-term longitudinal characteristic of these studies may not capture the potential psychological differences resulting from years of Brazilian jiu-jitsu engagement and technical development.

Based on the known positive psychological impact of combat sports and martial arts [8,9,10,15,21,24,26,27,28] combined with the limited number of investigations addressing the psychological differences resulting from Brazilian Jiu-jitsu practice [15,26,27,28], the selections of measuring tools related to performance and well-being were prompted. The novelty of this study is based on providing a comprehensive profile between athletes of different belt ranks. This study can serve as an initial exploration of Brazilian jiu-jitsu as a potential intervention to optimize the investigated psychological characteristics. Therefore, the purpose of the current study was to investigate the differences in mental strength, resilience, grit, self-efficacy, self-control, aggression, life satisfaction, and mental health disorders among Brazilian jiu-jitsu belt ranks. Based on previous findings, it was hypothesized that athletes in the higher belt rank group (e.g., black belts) would present higher mental strength, resilience, grit, self-efficacy, self-control, life satisfaction, and lower aggression and mental health disorders than athletes in the lower belt rank groups (e.g., white belts).

## 2. Materials and Methods

### 2.1. Participants

The current sample consisted of 420 Brazilian jiu-jitsu athletes, representing 331 males (78.8%) and 89 females (21.2%) from 18 to 60 years of age (mean age 38.2 ± 8.8). Participants included 121 white belts (28.8%), 118 blue belts (28.1%), 78 purple belts (18.6%), 46 brown belts (11%), and 57 black belts (13.6%). Of the total sample, 189 participants (45%) reported engaging in at least one Brazilian jiu-jitsu competition over the previous 12 months, and 231 participants (55%) reported no engagement with competitions during the same period. The sample included participants from forty-seven states and the District of Columbia. Nevada (19%), Pennsylvania (9%), and California (7.9%) were the states with the highest number of participants, and Rhode Island, West Virginia, and Wyoming had no representatives in this study.

### 2.2. Procedures

In the present study, data collection was conducted from April 19 to June 10, 2024, via Google Forms. Participants were recruited via a social media flyer linked to the online research form. The flyer was posted on Facebook and Instagram and distributed to Brazilian Jiu-Jitsu clubs across the United States. Participants included in the study consisted of males and females, ranging from 18 to 60 years, of all belt ranks (white, blue, purple, brown, and black), who were currently engaged in Brazilian jiu-jitsu classes for at least one class per week. Through the Google Forms link, participants answered to demographic (age, biological sex, and state of residency), and training-related questions (training experience, training frequency, training volume, belt rank, and competitive engagement over the previous 12 months), followed by the Mental Strength Scale, Brief Resilience Scale, Grit Scale, General Self-Efficacy Scale, Brief Self-Control Scale, Brief Aggression Questionnaire, Satisfaction with Life Scale, and Mental Health Disorders Screening Instrument for Athletes.

This study was anonymous, and no compensation was offered for participation. Prior to data collection, the present research received exempt status by the Liberty University Institutional Review Board as per 45 CFR 46:104(d): Category 2. (i)., clarifying that the data obtained by the author is recorded in a way that the identity of the human participants cannot readily be ascertained directly or through identifiers linked to the participants [33].

### 2.3. Materials

The 10-item Mental Strength Scale (MSS) was used to assess the athletes’ mental strength due to its good internal consistency reliability (Cronbach’s α = 0.80) [7]. Participants were asked to mark the best option (5-point Likert scale) about their thoughts over the past month while reflecting on their athletic engagement. Items 3, 6, 7, and 10 were positively worded, with a response of 1 representing “strongly disagree” and 5 representing “strongly agree.” Items 1, 2, 4, 5, 8, and 9 were negatively worded and reverse-coded. Total scores were determined by the average of the ten items, with 5 representing higher mental strength. The Cronbach’s alpha of the MSS in the current sample was 0.783.

The 6-item Brief Resilience Scale (BRS) was used to assess the athletes’ resilience due to its good internal consistency reliability (Cronbach’s alpha ranging from 0.80 to 0.91 in four studies) [34]. Participants were asked to mark one box per row (5-point Likert scale) that best describes them, with 1 representing “strongly disagree” and 5 representing “strongly agree”. Statements 1, 3, and 5 were positively worded, and 2, 4, and 6 negatively worded and reverse-coded. Total scores were determined by the average of the six items, with higher scores representing higher resilience. The Cronbach’s alpha of the BRS in the current sample was 0.82.

The 12-item Grit Scale (GS) was used to assess the athletes’ grit due to its good internal consistency reliability (Cronbach’s α = 0.85) [35]. Participants were asked to mark the statement (5-point Likert scale) that best described them compared to other people. Items 1, 4, 6, 9, 10, and 12 were answered on a scale from 1 for “not like me at all” to 5 for “very much like me.” Items 2, 3, 5, 7, 8, and 11 were reverse-coded. Questions represented two subscales, including consistency of interest (items 2, 3, 5, 7, 8, and 11) and perseverance of effort (items 1, 4, 6, 9, 10, and 12). The total scores were determined by the average of the 12 items, with higher scores representing higher grit. The Cronbach’s alpha of the GS in the current sample was 0.84.

The 10-item General Self-Efficacy Scale (GSES) was used to assess the athletes’ self-efficacy due to its good internal consistency reliability (Cronbach’s alpha between 0.76 and 0.90) [36]. Answers (4-point Likert scale) ranged from 1, representing “not at all true,” to 4, representing “exactly true.” Total scores were calculated by the sum of the 10 items, with higher scores indicating higher self-efficacy. The Cronbach’s alpha of the GSES in the current sample was 0.87.

The 13-item Brief Self-Control Scale (BSCS) was used to assess the athletes’ self-control due to its good internal consistency reliability (Cronbach’s alpha of 0.83 and 0.85 in two different studies) [37]. Participants were asked to select the statement that best represented them (5-point Likert scale), with answers ranging from 1 for “not at all like me” to 5 for “very much like me”. Statements 1, 6, 8, and 11 were positively worded, and statements 2, 3, 4, 5, 7, 9, 10, 12, and 13 were negatively worded and reverse-coded. Total scores were determined by the sum of the 13 items, with higher scores indicating higher self-control. The Cronbach’s alpha of the BSCS in the current sample was 0.84.

The 12-item Brief Aggression Questionnaire (BAQ) was selected to assess the athletes’ aggression due to its strong and significant reliability and brief characteristics [38,39]. Questions represent four subscales, including physical aggression (items 1, 2, and 3), anger (items 4, 5, and 6), verbal aggression (items 7, 8, and 9), and hostility (items 10, 11, and 12). Participants were asked to indicate (7-point Likert scale) the answer that best described them, ranging from 1 for “extremely uncharacteristic of me” to 7 for “extremely characteristic of me,” with item 4 reverse-coded. Total scores were calculated by the average of the 12 items, with higher scores indicating higher aggression. The Cronbach’s alpha of the BAQ in the current sample was 0.76.

The 5-item Satisfaction with Life Scale (SWLS) was selected to assess the athletes’ life satisfaction due to its good internal consistency reliability (Cronbach’s α = 0.87) [40]. Participants indicated (7-point Likert scale) the statement that best represented their thoughts ranging from 1 for “strongly disagree” to 7 for “strongly agree”. Total scores were calculated by the sum of the 5 items, with higher scores indicating higher life satisfaction. The Cronbach’s alpha of the SWLS in the current sample was 0.87.

The 14-item Mental Health Disorders Screening Instrument for Athletes (MHDSIA) was selected to assess the athletes’ degree of mental health disorders due to its good internal consistency reliability (Cronbach’s α 0.86) [41]. Participants were asked to select the number (7-point Likert scale) that best represented how often these items interfered with their lives outside of sports. Answers ranged from 1 for “never” to 7 for “always”. Total scores were determined by the sum of the 14 items, with higher scores indicating a higher degree of mental health disorders. The Cronbach’s alpha of the MHDSIA in the current sample was 0.82.

### 2.4. Statistical Analyses

In the current data, all normality assumptions were met. Primary analyses were conducted via independent *t*-tests to compare mental strength, resilience, grit, self-efficacy, self-control, aggression, mental health disorders, and life satisfaction between white and black belts. As secondary analyses, one-way ANOVA with Tukey’s post hoc tests were conducted to compare the dependent variables of mental strength, resilience, grit, self-efficacy, self-control, aggression, mental health disorders, and life satisfaction among belt ranks (white, blue, purple, brown, and black belts). Additionally, correlations between Brazilian jiu-jitsu training experience and each dependent variable were performed via Pearson’s *r*. Data were analyzed using IBM SPSS Statistics (Version 29) with an alpha level of 0.05.

## 3. Results

### 3.1. Demographic and Training Characteristics

Table 1 presents the participants’ demographic and training characteristics.

### 3.2. Group Comparison

Independent samples *t*-tests were conducted to investigate the differences in mental strength, resilience, grit, self-efficacy, self-control, aggression, life satisfaction, and mental health between white and black belts (Table 2). Moreover, one-way ANOVAs were performed to investigate the differences in the dependent variable among the five belt ranks (white, blue, purple, brown, and black).

For mental strength, independent samples *t*-tests revealed statistically significantly higher mental strength *t*(176) = −1.950, *p* = 0.026, Cohen’s *d* = −0.313 (small effect), 95% CI [−0.34303, 0.00210] in black belts than in white belts. However, ANOVA results revealed no statistically significant omnibus difference in mental strength among all Brazilian jiu-jitsu belt ranks *F*(4, 415) = 1.740, *p* = 0.140, η^2^ = 0.016.

For resilience, independent samples *t*-tests revealed statistically significantly higher resilience *t*(176) = −1.916, *p* = 0.028, Cohen’s *d* = −0.308 (small effect), 95% CI [−0.40633, 0.00596] in black belts than in white belts. However, ANOVA results revealed no statistically significant omnibus difference in resilience among all Brazilian jiu-jitsu belt ranks *F*(4, 415) = 1.955, *p* = 0.101, η^2^ = 0.018.

For grit, independent samples *t*-tests revealed statistically significantly higher total grit *t*(176) = −3.269, *p* < 0.001, Cohen’s *d* = −0.525 (moderate effect), 95% CI [−0.46761, −0.11557], ‘perseverance of effort’ *t*(176) = −3.417, *p* < 0.001, Cohen’s *d* = −0.549 (moderate effect), 95% CI [−0.48673, −0.13035], and ‘consistency of interest’ *t*(176) = −2.435, *p* = 0.008, Cohen’s *d* = −0.391 (small effect), 95% CI [−0.49726, −0.05201] in black belts than in white belts. ANOVA results revealed statistically significant omnibus differences in total grit among all Brazilian jiu-jitsu belt ranks *F*(4, 415) = 4.339, *p* = 0.002, η^2^ = 0.040. Tukey’s post hoc analyses showed significantly higher total grit in black belts than in white (*p* = 0.006) and blue belts (*p* = 0.020). No other post hoc comparisons were statistically significant (*p* > 0.05).

ANOVA results revealed statistically significant omnibus differences in ‘perseverance of effort’ among all Brazilian jiu-jitsu belt ranks *F*(4, 415) = 5.617, *p* < 0.001, η^2^ = 0.051. Tukey’s post hoc analyses showed significantly higher ‘perseverance of effort’ in black belts than in white (*p* = 0.004) and blue belts (*p* = 0.002). In addition, purple belts showed significantly higher ‘perseverance of effort’ than white (*p* = 0.029) and blue belts (*p* = 0.016). No other post hoc comparisons were statistically significant (*p* > 0.05). ANOVA results revealed no statistically significant omnibus differences in ‘consistency of interest’ among all Brazilian jiu-jitsu belt ranks *F*(4, 415) = 1.951, *p* = 0.101, η^2^ = 0.018.

For self-efficacy, independent samples *t*-tests revealed statistically significantly higher self-efficacy *t*(176) = −2.182, *p* = 0.015, Cohen’s *d* = −0.351 (small effect), 95% CI [−2.46319, −0.12373] in black belts than in white belts. However, ANOVA results revealed no statistically significant omnibus difference in self-efficacy among all Brazilian jiu-jitsu belt ranks *F*(4, 415) = 1.345, *p* = 0.252, η^2^ = 0.013.

For self-control, independent samples *t*-tests revealed statistically significantly higher self-control *t*(176) = −2.887, *p* = 0.002, Cohen’s *d* = −0.464 (small effect), 95% CI [−6.17424, −1.16054] in black belts than in white belts. However, ANOVA results revealed no statistically significant omnibus difference in self-control among all Brazilian jiu-jitsu belt ranks *F*(4, 415) = 2.082, *p* = 0.082, η^2^ = 0.020.

For aggression, independent samples *t*-tests revealed no statistically significant differences between black and white belts in total aggression *t*(176) = −0.292, *p* = 0.385, Cohen’s *d* = −0.047, 95% CI [−0.32500, 0.24132], physical aggression *t*(94.321) = −0.994, *p* = 0.161, Cohen’s *d* = −0.170, 95% CI [−0.82383, 0.27431], anger *t*(176) = 0.777, *p* = 0.219, Cohen’s *d* = 0.125, 95% CI [−0.22900, 0.52613], verbal aggression *t*(176) = −1.390, *p* = 0.083, Cohen’s *d* = −0.223, 95% CI [−0.71133, 0.12334], and hostility *t*(176) = 1.266, *p* = 0.104, Cohen’s *d* = 0.203, 95% CI [−0.14130, 0.64693]. ANOVA results revealed no statistically significant omnibus differences among all Brazilian jiu-jitsu belt ranks in total aggression *F*(4, 415) = 1.197, *p* = 0.312, η^2^ = 0.011, physical aggression *F*(4, 415) = 1.024, *p* = 0.394, η^2^ = 0.010, anger *F*(4, 415) = 2.024, *p* = 0.090, η^2^ = 0.019, verbal aggression *F*(4, 415) = 1.140, *p* = 0.337, η^2^ = 0.011, and hostility *F*(4, 415) = 1.052, *p* = 0.380, η^2^ = 0.010.

For life satisfaction, independent samples *t*-tests revealed statistically significantly higher life satisfaction *t*(176) = −2.718, *p* = 0.004, Cohen’s *d* = −0.437 (medium effect), 95% CI [−4.34459, −0.68919] in black belts than in white belts. ANOVA results revealed a statistically significant omnibus difference in life satisfaction among Brazilian jiu-jitsu belt ranks *F*(4, 415) = 2.533, *p* = 0.040, η^2^ = 0.024. However, no post hoc pairwise group comparisons were significant (*p* > 0.05).

For mental health disorders, independent samples *t*-tests revealed statistically significantly lower mental health disorders *t*(176) = 1.774, *p* = 0.039, Cohen’s *d* = 0.285 (small effect), 95% CI [−0.32333, 6.06205] in black belts than in white belts. ANOVA results revealed no statistically significant omnibus difference in mental health disorders among all Brazilian jiu-jitsu belt ranks *F*(4, 415) = 1.256, *p* = 0.287, η^2^ = 0.012.

### 3.3. Correlations

Table 3 presents the correlations between Brazilian jiu-jitsu training experience in years and the dependent variables. Significant positive correlations were found between Brazilian jiu-jitsu experience and mental strength, resilience, grit, consistency of interest, perseverance of effort, self-efficacy, self-control, aggression, physical aggression, verbal aggression, and life satisfaction.

## 4. Discussion

The current study investigated the differences in mental strength, resilience, grit, self-efficacy, self-control, aggression, life satisfaction, and mental health disorders among Brazilian jiu-jitsu belt ranks.

The mental strength results in the current study converge with previous findings, where more experienced combat sports athletes were more likely to present higher mental strength [7]. The differences in mental strength in Brazilian jiu-jitsu athletes are associated with the athletes’ skills and training experience, as seen through the significant differences in mental strength between white and black belts (small effect), and the significant correlation between training experience and mental strength (small effect).

Resilience results converge with previous studies that found a significant increase in resilience after 10–12 weeks of engagement with martial arts programs (non-Brazilian jiu-jitsu) in children, adolescents, and adults [8,10,42]. This difference in resilience in Brazilian jiu-jitsu athletes seems to depend on the athlete’s skill and training experience, as shown through the significant differences in resilience between white and black belts (small effect), and the significant correlation between training experience and resilience (small effect). These findings are supported by previous studies, which found a positive correlation between combat sports experience and resilience [29,30,31].

In the present study, significantly higher grit was found in black belts than in blue and white belts, which is similar to the findings by Lee et al. (2021), where taekwondo skills positively correlated with grit [12]. In children and adolescents, Sawyer et al. (2018) found that parent- and instructor-rated grit positively correlated with taekwondo skills [11]. The significant correlation found in the current study between training experience and grit (small effect) converges with Lorenco-Lima’s (2024) findings, which also showed a significant correlation between grit and combat sports experience (small effect) [13].

Significantly higher self-efficacy was found in black belts than in white belts. Similarly, Greco et al. (2019) reported a significant increase in self-efficacy (academic, emotional, and social) after 12 weeks of martial arts (non-Brazilian jiu-jitsu) training in adolescents [8]. Salchow et al. (2021) found significantly higher self-efficacy after six months of Kyusho Jitsu practice in breast cancer survivors [43]. Faro et al. (2020) explain that self-efficacy in experienced Brazilian jiu-jitsu athletes can be fundamental to sustaining focus and competitiveness, especially when fatigued [44]. In boxers, Chen et al. (2019) showed increased self-efficacy with a simultaneous increase in training experience, age, and competitive level. These findings converge with the present study, in which a significant correlation was found between training experience and self-efficacy (small effect) [32].

Significantly higher self-control was found in black belts than in white belts. Similarly, previous studies found significantly higher self-regulation and self-control after 4–6 months of martial arts classes with children and at-risk youth [45,46]. Blomqvist-Mickelsson (2019) found that five months of Brazilian jiu-jitsu training significantly increased self-control in adolescents and young adults [16]. Invernizzi et al. (2023) demonstrated that an acute Judo session effectively improved judokas’ inhibitory control and self-control ability [18]. In boxers, Chen et al. (2019) found self-control to increase simultaneously with advancements in age, competitive level, and training experience [32]. These findings converge with the present study, in which a significant correlation was found between training experience and self-control.

The results showed no significant differences in aggression among belt ranks. Kostorz and Sas-Nowosielski (2021) also found no difference in aggression based on combat sports and martial arts training experience and training rank in adolescents and adults [47]. The current results diverge from the findings of Blomqvist-Mickelsson (2019), who found decreased aggression after five months of Brazilian jiu-jitsu training in adolescents and young adults [15]. Wojdat and Ossowsky (2019) found a decrease in total aggression following a simultaneous increase in training experience, with significant differences observed in the first 2–3 years of Brazilian jiu-jitsu training [19]. Gorner et al. (2021) found that martial arts training experience negatively correlated (medium effect) with verbal aggression, suspicion, negativism, irritability, and assault [20]. Although the current study found no significant correlation between training experience and total aggression, positive correlations were found between training experience and the subscales physical aggression and verbal aggression. Although these positive correlations (small effect) may be perceived as negative, it seems reasonable to speculate that the training experience may encourage the athletes’ willingness to become physically and verbally aggressive if deemed necessary. Despite the lack of overall positive impact, the present study safeguards BJJ’s approach to the concern expressed by Lindell-Postigo et al. (2023), in which the non-educational and unreliable pedagogical nature of many combat sports could negatively affect their athletes [21].

Black belts reported higher life satisfaction than white belts. Similarly, Bai et al. (2023) found life satisfaction to be significantly improved, with a large effect, after 12 weeks of Tai Chi training [24]. On the other hand, Potoczny (2022) reported no direct effect of karate practice on life satisfaction [17]. However, an indirect association between karate practice and life satisfaction was found through reappraisal and self-control pathways. Rawat et al. (2022) discussed that, in Brazilian jiu-jitsu athletes, life satisfaction was found to be strongly associated with the length and quality of the athletes’ sports careers, supporting the current positive correlation between training experience and life satisfaction [23].

Black belts reported fewer mental health disorders than white belts. Similarly, short-term longitudinal studies have shown Brazilian jiu-jitsu’s effectiveness in improving mental health variables, including post-traumatic stress disorder [26,27] and psychopathological symptoms [26]. In children, 12 weeks of Brazilian jiu-jitsu training significantly decreased emotional symptoms, hyperactivity/inattention, the total difficulties score, and externalizing problems [28]. In the long term, Brazilian jiu-jitsu training provides opportunities for social engagement and support from like-minded individuals [26]. Farrer (2019) and Sugden (2021) suggest that much of Brazilian jiu-jitsu’s benefits are drawn from the social environment and the emphasis on self-development and personal growth [48,49].

This research is not free of limitations. First, its cross-sectional nature prevents any causality assumptions or the observation of changes over time. Between-subject differences were observed rather than changes within subjects, providing initial but limited evidence of non-causal long-term effects of Brazilian jiu-jitsu training. Second, the reliance on self-reported answers may have led to social desirability bias, despite the attempt to mitigate the issue via anonymous data collection. It is possible that responses reflected the participants’ implicit attitudes or personal desires and objectives toward the situations asked in the questions. Third, regardless of the inclusion of a state-diverse group with representatives from 47 states and the District of Columbia, self-selection bias led to large variability in representativeness among states, potentially limiting the generalizability of the study. Fourth, the small number of studies investigating the psychological aspects of Brazilian jiu-jitsu training limited the theoretical foundation and discussion of the current research. Fifth, due to the diverging results in the *t*-tests and ANOVA in the analyses between white and black belts, caution should be taken when interpreting the results. Although the current findings are suggestive of differences between white and black belts, further analyses are necessary to determine whether the rank effect remains after controlling for the athletes’ age. Future studies are encouraged to explore whether these characteristics are prerequisites for continued Brazilian jiu-jitsu training or whether a causal relationship exists.

In conclusion, the present study suggests that black belts presented significantly higher mental strength, resilience, grit, self-efficacy, self-control, and life satisfaction, and lower mental health disorders than white belts. Moreover, black belts presented higher grit than blue belts. More experienced Brazilian jiu-jitsu athletes were more likely to present higher mental strength, resilience, grit, self-efficacy, self-control, total aggression (physical and verbal aggression) and life satisfaction than less experienced athletes. Despite the positive correlations (small effect) found in total aggression, physical aggression, and verbal aggression, no significant positive or negative differences were observed in aggression among belt ranks.

## Figures and Tables

**Table 1 jfmk-10-00100-t001:** Participants’ demographic and training characteristics.

	White (*N* = 121)	Blue (*N* = 118)	Purple (*N* = 78)	Brown (*N* = 46)	Black (*N* = 57)
**Demographic**					
Age (SD)	36.18 (9.27)	36.92 (8.43)	39.09 (8.49)	40.11 (7.54)	42.44 (7.93)
Male n (%)	92 (76.03)	93 (78.81)	59 (75.64)	37 (80.43)	50 (87.72)
Female n (%)	29 (23.97)	25 (21.19)	19 (24.36)	9 (19.57)	7 (12.28)
**Training**					
Experience (SD)	1.88 (2.51)	4.25 (2.89)	7.37 (3.42)	10.17 (3.09)	16.53 (6.05)
Days/Week (SD)	3.06 (1.17)	3.55 (1.30)	3.69 (1.14)	4.04 (1.53)	4.30 (1.49)
Hours/Week (SD)	4.81 (3.77)	5.72 (3.32)	6.47 (3.52)	7.41 (3.92)	8.79 (5.17)
Competitions (SD)	0.70 (1.08)	1.45 (2.49)	1.23 (1.94)	1.22 (2.01)	0.93 (2.03)

Note: The statistics presented are means except where explicitly indicated; Experience: represents the participants’ training experience in years; Competitions: represents the number of competitions engaged by the participants over the previous 12 months.

**Table 2 jfmk-10-00100-t002:** Belt rank group comparison means and standard deviations.

	White (*N* = 121)	Blue (*N* = 118)	Purple (*N* = 78)	Brown (*N* = 46)	Black (*N* = 57)
Mental Strength	4.02 (0.54) *	3.99 (0.48)	4.04 (0.46)	4.11 (0.47)	4.19 (0.55) *
Resilience	3.69 (0.63) *	3.66 (0.65)	3.84 (0.61)	3.74 (0.61)	3.89 (0.69) *
Grit	3.70 (0.57) ^a,^*	3.73 (0.55) ^b^	3.91 (0.49)	3.84 (0.46)	3.99 (0.52) ^a,b,^*
Perseverance of effort	4.02 (0.59) ^a,b,^*	4.01 (0.57) ^c,d^	4.26 (0.48) ^a,c^	4.14 (0.50)	4.33 (0.49) ^b,d,^*
Consistency of interest	3.38 (0.71) *	3.45 (0.72)	3.57 (0.67)	3.54 (0.67)	3.65 (0.69) *
Self-Efficacy	33.64 (3.79) *	34.26 (3.82)	34.44 (3.68)	34.13 (3.48)	34.93 (3.46) *
Self-Control	45.30 (7.47) ^a,^*	46.65 (8.45)	46.91 (7.66)	46.33 (7.89)	48.96 (8.78) ^a,^*
Aggression	3.12 (0.83)	2.93 (0.89)	3.05 (0.91)	3.17 (0.83)	3.16 (1.02)
Physical aggression	3.43 (1.52)	3.30 (1.64)	3.30 (1.52)	3.68 (1.61)	3.70 (1.81)
Anger	2.29 (1.22)	1.90 (0.90)	2.13 (1.15)	2.06 (0.95)	2.15 (1.12)
Verbal aggression	3.79 (1.33)	3.85 (1.33)	3.96 (1.23)	4.19 (1.22)	4.08 (1.30)
Hostility	2.98 (1.24)	2.66 (1.24)	2.81 (1.36)	2.75 (1.22)	2.73 (1.26)
Life Satisfaction	25.13 (5.80) *	25.74 (6.86)	27.08 (5.51)	26.91 (5.38)	27.65 (5.70) *
Mental Health Disorders	35.41 (9.68) *	32.97 (9.91)	33.15 (10.99)	33.46 (9.74)	32.54 (10.18) *

Note: ^a,b,c,d^ *p* < 0.05 via ANOVA; * *p* < 0.05 via independent samples *t*-test; equal letters/characters represent differences between groups; Mean (SD).

**Table 3 jfmk-10-00100-t003:** Correlations between training experience and dependent psychological variables.

	*r*	*p*
Mental Strength	0.113	0.010 *
Resilience	0.144	0.002 *
Grit	0.179	0.000 *
Consistency of Interest	0.123	0.006 *
Perseverance of Effort	0.193	0.000 *
Self-Efficacy	0.123	0.006 *
Self-Control	0.090	0.033 *
Aggression	0.112	0.011 *
Physical Aggression	0.144	0.002 *
Anger	0.011	0.409
Verbal Aggression	0.155	0.001 *
Hostility	−0.037	0.226
Life Satisfaction	0.126	0.005 *
Mental Health Disorders	−0.011	0.408

Note: * *p* < 0.05.

## Data Availability

The data presented in this study are available upon request from the corresponding author.

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
