# Peer review of "Rank-Based Psychological Characteristics in Brazilian Jiu-Jitsu Athletes: Mental Strength, Resilience, Grit, Self-Efficacy, Self-Control, Aggression, Life Satisfaction, and Mental Health"

_jfmk, 2025, doi:10.3390/jfmk10020100_

Round 1

Reviewer 1 Report

Comments and Suggestions for Authors

The study is relevant in modern sports science and may have a significant impact on the understanding of the benefits of sport not only physically but also psychologically. The study may also be useful for other sports - although the study relates to Jiu-Jitsu athletes, its findings may also be applicable to other martial arts or other sports, especially when it comes to the psychological well-being and behavior of athletes. 

However, I would like to make some comments:

1.      A detailed analysis of related studies has been done in the discussion section, but the introduction provides poor overview of research related to the psychological characteristics (Mental Strength, Resilience, Grit, Self-Efficacy, Self-Control, Aggression, Life Satisfaction, and Mental Health) of the study.  The introduction should be supplemented.

2.      It is unclear why these specific research objects (Mental Strength, Resilience, Grit, Self-Efficacy, Self-Control, Aggression, Life Satisfaction, and Mental Health) were chosen for the study. Why, for example, is aggressiveness analyzed alongside mental characteristics? A more detailed explanation would be needed.

3.      The novelty of the study is not sufficiently substantiated.

4.      The third table is not commented.

5.      Interesting  why the data was not analyzed by age, since the age range of the respondents is very wide, e.g. people of different age groups may experience different mental health problems.

Author Response

Comment 1: [A detailed analysis of related studies has been done in the discussion section, but the introduction provides poor overview of research related to the psychological characteristics (Mental Strength, Resilience, Grit, Self-Efficacy, Self-Control, Aggression, Life Satisfaction, and Mental Health) of the study.  The introduction should be supplemented.]

Response1: [Introduction restructured.]

Comment 2: [It is unclear why these specific research objects (Mental Strength, Resilience, Grit, Self-Efficacy, Self-Control, Aggression, Life Satisfaction, and Mental Health) were chosen for the study. Why, for example, is aggressiveness analyzed alongside mental characteristics? A more detailed explanation would be needed.]

Response 2: [Comment addressed in the introduction. These variables were selected due to their relevance to the practice of Brazilian jiu-jitsu based and based on previous studies with other combat sports and martial arts.]

Comment 3: [The novelty of the study is not sufficiently substantiated.]

Response 3: [Novelty addressed in the introduction].

Comment 4: [The third table is not commented.]

Response 4: [Comment added.]

Comment 5: [Interesting why the data was not analyzed by age, since the age range of the respondents is very wide, e.g. people of different age groups may experience different mental health problems.]

Comment 5: [Recommendations made for future studies to address this matter. Although valuable for future studies, the requested new analysis would result in a different aim than the proposed for this study. This first of a kind study has the intent to reveal any group differences and that's why ANOVA and t-test were presented. The idea was not to isolate the belt effect but present the profile of different belt ranks. This initial study could serve as the foundation for future studies.]

Reviewer 2 Report

Comments and Suggestions for Authors

Thank you for the opportunity to read this manuscript. The authors have described a cross-sectional study of the association between skill level in Brazilian Jiu-Jitsu (BJJ) and various psychological/emotional characteristics. A good sample of athletes has been obtained and the coverage across the US is impressive.  I do not have any particular problem with how the study was conducted, however, in my view, the presentation of ideas and justification of the research needs improvement before publication. I also think some further consideration of the analysis is necessary.    Introduction

  1. I didn't really follow the train of thought presented in the introduction. There is discussion of the challenges of BJJ, the fact that personal psychology influences persistence, some discussion of Social Learning Theory, the value of BJJ for those with post-traumatic stress, and the purpose of the study. I found it a bit random and think some more effort is needed to create a clear broad-to-narrow flow of ideas.
  2. Bandura's theory seems somewhat unrelated to the specific matter at hand. Are there better theories to consider - e.g., that make some relevant predictions (i.e., facilitate the hypothesis)? Or if not, is a theoretical focus important? If theory didn't really drive this research, why not spend the words explaining and motivating the study in more detail?
  3. I would like to see more justification for the study. Why is studying the association between BJJ rank differences and these psychological characteristics interesting? How does it help us to know that someone who has passed more BJJ milestones possesses certain psychological characteristics? Why are these particular characteristics being studied?
  4. Related to the previous point, the constructs of interest are not described (e.g., grit), and I think need some introduction. 

Method/Results

  1. There is scope here for further analyses here - e.g., regression models that would tell us if belt effects remain after accounting for age or experience (I think this would be interesting for the readers). I encourage further analyses, or some acknowledgment of this as future research. 
  2. T-tests are done between black and white belts. An ANOVA is also done with post-hoc tests (including between black and white belts). I see a few issues here:
    1. There is a duplication of analysis which doesn't make much sense.
    2. What to do when the results contradict each other? What if there is a significant t-test and a non-significant ANOVA (e.g., self-control) or what if there is a significant t-test, a significant ANOVA, but the black vs. white belt post hoc test is non-significant (e.g., life satisfaction). Disagreement between results is not a good sign for a robust effect and I think some caution should be offered to interpret these cases.
    3. If the black vs. white belt test is already done as a t-test, then there is no need to do it after the ANOVA and the omission of this post-hoc test should increase power to detect significant results.
    4. Why not just do the ANOVA and post hoc tests including the black vs white belt test, and leave out the t-test? Or as per above, a regression model could be done for each measure that would also deliver the same information parsimoniously, and consider age and experience at the same time.  
  3. There is inconsistency in the number of decimal places reported (e.g., CIs reported to 5 decimal places).

Discussion

  1. In my opinion, much literature review done in the discussion is better placed in the introduction. In this discussion, we learn in some detail about research with athletes that could have served to motivate the current study and explain the hypotheses (e.g., research from other sports suggests that more skilled athletes have higher grit, hence it is hypothesised that grit should be positively associated with belt grade).
  2. I think that the focus of this discussion should be on why the results are as they are. Why do black belts have higher life satisfaction? Why do they report fewer mental health disorders? In my view, the bulk of the discussion should be about this.
  3. I think that some future research should be suggested - what are the next steps?

Author Response

Introduction

Comment 1: [I didn't really follow the train of thought presented in the introduction. There is discussion of the challenges of BJJ, the fact that personal psychology influences persistence, some discussion of Social Learning Theory, the value of BJJ for those with post-traumatic stress, and the purpose of the study. I found it a bit random and think some more effort is needed to create a clear broad-to-narrow flow of ideas.]

Response 1: [ Introduction restructured for clarity.]

Comment 2: [Bandura's theory seems somewhat unrelated to the specific matter at hand. Are there better theories to consider - e.g., that make some relevant predictions (i.e., facilitate the hypothesis)? Or if not, is a theoretical focus important? If theory didn't really drive this research, why not spend the words explaining and motivating the study in more detail?]

Response 2: [Motivation added.]

Comment 3: [I would like to see more justification for the study. Why is studying the association between BJJ rank differences and these psychological characteristics interesting? How does it help us to know that someone who has passed more BJJ milestones possesses certain psychological characteristics? Why are these particular characteristics being studied?]

Response 3: [Justification added.]

Comment 4: [Related to the previous point, the constructs of interest are not described (e.g., grit), and I think need some introduction.]

Response 4: [Describing all eight investigated constructs and subscales would increase the total number of words. We addressed these constructs in the methods section to a limited extent to fulfill the journal's requirements.]

Methods and Results

Comments 1-4: [After consideration we decided to appeal this comments. We added a recommendation for future studies to control for age. Although valuable for future studies, the requested new analyses would result in a different aim than the proposed for this study. This first of a kind study has the intent to reveal any group differences and that's why ANOVA and t-test were presented. The idea was not to isolate the belt effect but present the profile of different belt ranks. This initial study could serve as the foundation for future studies.]

Discussion

Comment 1: [In my opinion, much literature review done in the discussion is better placed in the introduction. In this discussion, we learn in some detail about research with athletes that could have served to motivate the current study and explain the hypotheses (e.g., research from other sports suggests that more skilled athletes have higher grit, hence it is hypothesized that grit should be positively associated with belt grade).]

Response 1: [Introduction modified as suggested.]

Comment 2: [I think that the focus of this discussion should be on why the results are as they are. Why do black belts have higher life satisfaction? Why do they report fewer mental health disorders? In my view, the bulk of the discussion should be about this.]

Response 2: [At this point we have no definite results to explain why the results are the way they are. We used the discussion section to contrast and compare the current results with previous findings.]

Comment 3: [I think that some future research should be suggested - what are the next steps?]

Response 3: [Suggestion added to discussion.]

Round 2

Reviewer 2 Report

Comments and Suggestions for Authors

I would like to thank the authors for their changes which I think have improved the manuscript. While I like the study, I still have some reservations about how the results have been done. I accept the author's comments about not doing more extensive analyses but remain confused about the choices made in the current analysis (in particular the use of t-test as the primary analysis in a study of multiple belt ranks). Caution has been acknowledged but, in my opinion, this is insufficient. 

Introduction:

Page 2. "Moreover, the shot-term longitudinal characteristic of these studies may not capture the potential psychological differences resulting from years of Brazilian jiu-jitsu engagement and technical development."

It is unclear which studies are being referred to - perhaps the sentence just needs to be rephrased and moved closer to the references in question.

I still think better motivation is needed - what is the pragmatic value of doing this work? It is clear that the broad idea is to "to better understand the potential differences in psychological characteristics among Brazilian jiu-jitsu athletes of different belt ranks." But why? To understand what makes a good BJJ athlete? To do an initial assessment of the idea the BJJ is a good approach to mental health improvement? Both? Other things?

Methods:

"The 12-item Brief Aggression Questionnaire (BAQ) was selected to assess the 
athletes’ aggression due to its strong and significant test-retest reliability, varying from .68 to .80 among the four subscales, and going as high as .81 for the total BAQ [38,39]. 

This justification doesn't make much sense because the study doesn't "test-retest". Perhaps the idea is to indicate that it is broadly speaking a reliable measure, but it would be better to justify it another way as well - even if it is just that is commonly used by researchers, appropriately short etc. 

Results:

The standard practice is to run the omnibus ANOVA first, and if no significant difference is found, stop there. If there is a significant difference, then look for differences with post-hoc tests. I know that standard practice doesn't always apply, but running the t-tests independent of what the ANOVA says is not really justified in this manuscript. In my view, ANOVA followed by t-tests fits the hypotheses as they have been setup - an approach that will also test the black belt vs. white belt difference given an overall result. Why is the t-test the primary analysis?

There is inconsistency in the decimal places reported

Discussion:

I don't see the need for the variance explained statistics in the discussion - I think this should go in the results or be removed.

Page 7: "Mental strength results in the current study converge with previous findings where combat sports experience positively influenced the athletes’ mental strength"

We don't know for sure if mental strength grew with experience, or if people with mental strength are likely to persist in the sport. 

Author Response

Comment 1: [It is unclear which studies are being referred to - perhaps the sentence just needs to be rephrased and moved closer to the references in question.]

Response 1: [Sentence reformulated and references added.]

Comment 2: [I still think better motivation is needed - what is the pragmatic value of doing this work? It is clear that the broad idea is to "to better understand the potential differences in psychological characteristics among Brazilian jiu-jitsu athletes of different belt ranks." But why? To understand what makes a good BJJ athlete? To do an initial assessment of the idea the BJJ is a good approach to mental health improvement? Both? Other things?]

Response 2: [Suggestion added.]

Comment 3: [This justification doesn't make much sense because the study doesn't "test-retest". Perhaps the idea is to indicate that it is broadly speaking a reliable measure, but it would be better to justify it another way as well - even if it is just that is commonly used by researchers, appropriately short etc.]

Response 3: [Questionnaire chosen based on its good reliability and brief characteristic. Text modified.]

Comment 4: [The standard practice is to run the omnibus ANOVA first, and if no significant difference is found, stop there. If there is a significant difference, then look for differences with post-hoc tests. I know that standard practice doesn't always apply, but running the t-tests independent of what the ANOVA says is not really justified in this manuscript. In my view, ANOVA followed by t-tests fits the hypotheses as they have been setup - an approach that will also test the black belt vs. white belt difference given an overall result. Why is the t-test the primary analysis?]

Response 4: [The initial and main idea of this article was to compare white and black belts as presented in our hypotheses. There seems to be a higher consensus between the standard skill level of white belts and the requirements to achieve a black belt. The variability and lack of consensus in the middle belts (blue, purple, and brown) is also seen in the DVs assessed. Our main findings are resulting from the t-tests as the ANOVAs were not sensitive enough to detect many of the differences between white vs. black belts. We added a sentence under statistical analyses reflecting this explanation. Would it be satisfactory the removal of the ANOVA results and middle belt data?"

Comment 5: [I don't see the need for the variance explained statistics in the discussion - I think this should go in the results or be removed.]

Response 5: [Variance removed].

Comment 6: [Page 7: "Mental strength results in the current study converge with previous findings where combat sports experience positively influenced the athletes’ mental strength."]

Response 6: [Sentence reformulated.]

Round 3

Reviewer 2 Report

Comments and Suggestions for Authors

Thank you to the authors for the changes made.

I remain unconvinced by the analysis (and don't see a problem presenting a more conservative analysis using all the data, then focusing on black and white belts). However, I will defer to the editor or any other reviewers from here. In addition, a personal communication from the first author, in my view, is not an appropriate justification. This should be replaced with another reference or removed. 

Kind regards

Author Response

Comment 1: [I remain unconvinced by the analysis (and don't see a problem presenting a more conservative analysis using all the data, then focusing on black and white belts).]

Response 1: [We understand our choices may diverge from the usual practices, but we deeply believe in their relevance of t-test findings to Brazilian jiu-jitsu coaches and practitioners involved in the field. By only keeping the ANOVA results, we would remove several important (in our view) findings. We aim to present all significant differences and further explore the various reasons in future studies.]

Comment 2: [In addition, a personal communication from the first author, in my view, is not an appropriate justification. This should be replaced with another reference or removed.]   Response 2: [Personal communication removed, since no reference is available to replace it at this time.]